# Chiral Separations of Pyrethroic Acids Using Cyclodextrin Selectors

**DOI:** 10.3390/molecules27248718

**Published:** 2022-12-09

**Authors:** Zoltán Juvancz, Rita Bodáné-Kendrovics, Zita Laczkó, Róbert Iványi, Erzsébet Varga

**Affiliations:** 1Rejtőff Sándor Faculty of Light Industry and Environmental Engineering, Institute of Environmental Engineering and Natural Science, Óbuda University, Doberdó út 6, H-1034 Budapest, Hungary; 2Cyclolab Ltd., Illatos út 7, H-1097 Budapest, Hungary

**Keywords:** pyrethroic acids, chiral separations, gas chromatography, supercritical fluid chromatography, capillary electrophoresis, chiral selectivity–structure relationships

## Abstract

Pyrethroid insecticides are broadly used. They have low toxicity for warm-blooded living creatures, but high toxicity for both insects and fish. Therefore, it is important to reduce the environmental impact of pyrethroids. Pyrethroic acids are chiral compounds. An effective way to decrease pollution is to use enantio-pure insecticide products instead of their racemic mixtures. Enantiomer-pure products require enantiomer selective synthesis and analysis. The chiral selective analysis of pyrethroic acids (an intermediate of pyrethroids) is also important in terms of process control and from the point of view of their degradation metabolism in the environment. This study used various enantiomeric selective chromatographic methods for the separation of different pyrethroic acids, including gas chromatography, supercritical fluid chromatography and capillary electrophoresis. Systematic experiments were conducted to find the optimum conditions for their chiral separation. The employed enantio-selective agents were cyclodextrin derivatives with different ring sizes and substitution patterns. The β-cyclodextrin proved to be excellent for the chiral separation of these acids. The different chiral recognition mechanisms were established using different ring-sized cyclodextrins. The results of these systematic studies demonstrated the correlations of the chiral selectivity features of selectors and the structures of analytes.

## 1. Introduction

### 1.1. The Phenomenon of Chirality

Chiral molecules are asymmetric molecules that are not identical to their mirror images [1]. A chiral molecule and its mirror image molecule are called enantiomers, or an enantiomeric pair.

The properties of the members of an enantiomer pair are the same in a homogeneous (achiral) environment. However, they show different properties in an inhomogeneous, chiral environment (e.g., a magnetic field, receptor sites and interactions with other asymmetric molecules). The molecules of living organisms (e.g., amino acids, sugars, DNA, receptors, etc.) are dominantly asymmetric and several inorganic molecules are also chiral (e.g., quartz crystals). The members of an enantiomeric pair can show rather different biological effects in several cases. For example, every essential amino acid has only one member (L) of their enantiomeric pair in the proteins.

The importance of chiral-pure pharmaceuticals is evident following the Contergan scandal [2]. The pharmaceutical medicine Contergan (thalidomide) was a mixture of enantiomer molecules, in which the (*R*) enantiomer produced a desirable antiemetic effect. The other isomer, the (*S*)-enantiomer of the enantiomeric pair, was toxic and produced teratogenicity side effects. More than 10,000 children were born with serious deformations (phocomelia) when their mothers had taken Contergan during their pregnancy. Due to the Contergan scandal, the regulatory authorities have only allowed the introduction of enantiomer-pure medication products recently [3]. A chiral drug can contain less than 0.1% of its optical isomer. The application of enantiomer-pure agrochemicals is progressively gaining more market share [4]. Applying chiral-pure agrochemicals has become more accepted because the members of an enantiomer pair can have different properties in terms of their use and toxic effect in nature. There have been several examples of their different effects (pyrethrins, conazole, lindane, phenoxy propionic acids, metalaxyl, metolachlor, etc.).

### 1.2. Pyrethroids

Pyrethroids are broadly used natural and synthetic insecticides [5]. The natural product, pyrethrum, is extracted from chrysanthemum flowers. Synthetic products have more intensive insecticidal effects than their natural analogs. There are many synthetic pyrethroids, most of which are esters. Generally, the acidic units are pyrethroic acids (Figure 1). The acidic elements usually have two chiral centers.

The diastereomers of pyrethroic acids are called *cis*/*trans* isomers according to the relative configuration of the carboxylic and substituted vinyl (e.g., dimethyl-, dichloro-, dibromo-vinyl) groups on the cyclopropane ring. These acids are frequently esterified with optically active alcohols (cypermethrin, cyfluthrin, and deltamethrin), creating additional stereoisomers of the insecticidal products. The structural isomers of pyrethroids have different biological effects [6,7,8,9,10]. Pyrethroids have low toxicity for mammals, but high toxicity for bees and fish [6,7]. They have neurotoxic effects through the sodium channel [7].

The (1*R*) *cis* isomers of acids show much more effective insecticidal activity than the other isomers. For example, the deltamethrin pesticide has three chiral centers, but only one of the eight isomers (α*S*,1*R*,3*R*′) can act as a good and effective pesticide. On the other hand, every isomer has toxic side effects in fish [11]. A certain amount of enantiomeric-pure deltamethrin causes the same useful effect as eight times the amount of the stereoisomeric mixture of deltamethrin. The enantiomeric-pure product’s side effects are eight times less concerning the environmental impact than with a mixture of all the stereoisomers, showing the same useful effects. For this reason, enantiomeric-pure (α*S*,1*R*,3*R*′) deltamethrin has already been introduced to the market.

The significant difference in the biological activity of these enantiomers requires chiral selective syntheses [4,12], and highly efficient chiral analyses with high-performance liquid chromatography (HPLC) [6,8,13], supercritical fluid chromatography (SFC) [14,15] and gas chromatography (GC) [16]. The enantiomer selective analyses of pyrethroic acids are also important from the point of view of production processes and metabolism control in the environment. The GC [17], SFC [18], or capillary electrophoresis (CE) methods [19,20,21] are very useful in the chiral analysis of pyrethroic acids. Several chiral separations, however, remain unsolved in the field, particularly in the production of pyrethroids.

### 1.3. Separation of Enantiomers

There is no general chiral selector for every enantiomeric pair [22]. Chiral chromatographic separations require exact matches between the interaction groups of the selector and interaction groups of analytes [23,24]. Namely, the three-point interaction recognition mechanism requires appropriate geometrical arrangements and chemical compatibility among the interacting groups of the selectors and selectants. According to the simplified three-point interaction model, one of the enantiomers interacts with three interacting groups (points) with the selector. The other enantiomer can interact with only two interacting groups (points) of the selectors. The three interactions create a more stable association with the selector than with the two interactions. These interaction differences in the chiral recognition mechanism are the basis of chiral selective chromatography. The isomers with three-point interactions show longer retention times than isomers with only two-point interactions in GC. A rigid chiral selector can separate rigid analytes with high selectivity but can only separate a narrow band of the analytes [25]. A selector with moderate rigidity can separate a broad spectrum of the analytes, but with moderate selectivity.

The rigid structure of the cyclopropane ring of pyrethroic acids offers high enantiomer selectivity even with moderately rigid cyclodextrin-based chiral selectors [26].

### 1.4. Cyclodextrins

Cyclodextrins (CDs) are the most frequently used chiral selectors in capillary column separations [25,27,28]. Moderately rigid cyclodextrin selectors can appropriately separate a broad range of enantiomers if the used chromatographic technics offer high efficiency (e.g., a capillary column using GC and CE) [25].

CDs are cyclic oligosaccharide molecules [29], where the glucose units join with α-1,4-glycosidic linkages. The most important members of CDs consist of six, seven, or eight D (+)-glucopyranose units, which are assigned α, β, and γ Greek letters, respectively. The chemical structures of β-CD are shown in Figure 2. CDs have a cavity inside of the ring where they can hold molecules (guests) as hosts in their cavities. The strength of inclusion depends on the size, chemical character, shape, and chirality of the guest molecules.

The most important character of CDs is their excellent chiral recognition ability for the extremely broad spectra of enantiomers in analytical chemistry. CD molecules are the most popular chiral selectors of capillary columns in GC, SFC, and CE. Enantiomers, with any functional groups or structure, can be separated with CD-based selectors [17,24,25,26,27,28,29,30,31]. CDs have broad chiral selectivity spectra due to the following reasons.

CDs have numerous, non-uniform chiral centers (35 in β-CD). CDs have twisted truncated cone shapes instead of a regular, symmetrical cone (Figure 3). Chemically, the glucose units repeatedly connect to one another in the ring of the CD, but they are different sterically. The lengths and directions of the bonds around the chiral centers are different, caused by their twisted shapes [25]. The derivatized CDs show a more distorted shape than their native compounds, and the distortions increase with the volume of the substituent groups.

The hydroxyl groups of CDs can be substituted by different functional groups (e.g., methyl, phosphate, sulfate, amino, naphthyl and acetate) to add extra interaction abilities to the interaction potentials of underivatized, native CDs [29]. Some substituent groups (such as hydroxypropyl and naphthyl-ethyl carbamoyl) add an extra chiral center to CDs, further broadening their recognition spectra [25]. The substituents not only offer different interaction sizes, but also modify the cones’ shape.

Most CD derivatives are randomly substituted products and are mixtures of a large number of isomers [29]. They differ in the numbers and the positions of the substituents and almost every isomer has different chiral recognition features. Moreover, the substituents can be different in their chemical structures and arrangements in a derivatized CD [28].

The derivatized CDs have a rather flexible structure. CDs can change their shape to interact with analytes using the “induced fit” mechanism, broadening their selectivity spectra [29].

The ionizable CDs can change their selectivity spectra depending on their neutral or ionized states [32]. They can show different chiral recognition properties for analytes according to their ionized states. Moreover, their chiral recognition properties are also influenced by the ionized states of the analytes.

The types of background buffer or mobile phases (normal, reverse and polar-organic) determine the chiral selectivity features of CDs [25].

CDs can separate enantiomers, with carbon atom centers, and also phosphorus, nitrogen, or sulfur atoms in the chiral centers.

CDs can also separate chiral molecules with planar or axial chirality [30].

CDs have moderate rigidity; therefore, they generally produce lower chiral selectivity values than rigid selectors (e.g., amino acids, cellulose, etc.). Their relatively low chiral selectivity values are often not enough for baseline separation on moderately efficient packed columns. This is why the packed column techniques use CD chiral selectors less frequently than cellulose and amylose-based chiral selectors, which can show high selectivity values. On the other hand, CDs are the most popular chiral selectors, useing capillary columns with high-efficiency values [25]. Several hundred thousand theoretical plate efficiency values of these techniques offer baseline resolutions even for enantiomer pairs with only 1.01 chiral selectivity values.

Finding an appropriate chiral selector is frequently the result of a trial-and-error process. However, certain rules can be established for the chiral recognition conditions with systematic studies using model compounds. Pyrethroic acids have rather rigid structures [26]; therefore, the chemical structure vs chiral selectivity trends can be recognized in line with our expectations.

This study presents systematic research to show the chiral selectivity structure relationships using pyrethroic acids. In our research, we used different-sized cyclodextrins (α, β, γ) for chiral separations of various pyrethroic acids. The variously substituted CDs were applied (permethylated, partially methylated, amino-substituted etc.) as chiral selectors to determine the best fitting interacting groups for their chiral recognitions. The different derivatives of pyrethroic acids (free acids, methyl esters, etc.) were also tested to find the best form of pyrethroic acids for their chiral recognitions. We applied the following capillary chromatographic techniques: GC, SFC, and CE.

Our conclusions (structure–chiral selectivity correlations) can be used to solve several other chiral separations faster than using the trial-and-error method.

## 2. Result and Discussions

### 2.1. Results of Gas Chromatographic Measurements

The GC methods proved to be excellent for the chiral separation of pyrethroic acids [17]. Even the first capillary gas chromatographic chiral separation procedure was performed on a cyclodextrin chiral selector using a *cis*-permethrinic methyl ester enantiomeric pair [33]. This research used molten permethyl β-CD in a supercooled state as the chiral selector. The efficiency of this column was moderate in a supercooled state. However, the efficiency of the further columns improved significantly using a silicone polymer matrix with the same solved permethyl β-CD selector [17,18]. The 9400 theoretical plates/meter were obtained on a 50 µm diameter column using chemically bonded permethyl β-CD that contained a silicone polymer stationary phase (Chirasil-Dex) [18]. The silicone polymeric matrices offer high efficiency due to their low resistance to mass transfer. The chiral selectivity values of the silicone polymer that contained cyclodextrin chiral stationary phases did not decrease much compared to the molten selector selectivity values. Namely, the selectivity values show a level-up curves in the function of the concentration of the chiral selector in the polymeric silicone matrix [34].

In this study, every tested enantiomer of pyrethroic acids could be successfully separated using GC methods. The measurements were performed at different temperatures (60–180 °C) because the boiling points, inclusion properties, and chiral recognition features of the tested compounds are rather different.

Four chiral selective stationary phases were tested to find the ideal ring size of cyclodextrins for separating pyrethroic acids. These were Alfa Dex 120 (a mixture of silicone polymer and permethyl α-CD, Supelco), Chirasil-Dex (an in-house-made silicone polymer containing chemically bonded permethylated β-CD [18]), Cydex B (a mixture of silicone polymer and permethyl β, SGE), and Gamma Dex 120 (a mixture of silicone polymer and permethyl γ-CD, Supelco).

Every tested compound was measured at least at three different temperatures. The linearity of the natural logarithm of the chiral selectivity values (lnα) in the function of the inverse of absolute temperature (1/T) was tested. If the ln α-1/T relation is linear for the given enantiomeric pair, only one chiral recognition mechanism is responsible for this chiral separation [30]. The linearity of the curves was checked with regression calculations, as presented in Figure 4. The regression values (R^2^) were at least 0.94 in every case. The linearity of these regression calculations proved that the separation processes belong to only one chiral recognition mechanism for a given enantiomeric pair. This linearity allows the calculations of selectivity values at 100 °C, from retention times measured at different temperatures. Cydex-B showed very low selectivity for the *trans-*permethrinic acid methyl ester. Therefore, its values are not marked in Figure 4.

For the sake of comparison, the calculated selectivity values are summarized for 100 °C (Table 1). The measurements were performed or calculated for this temperature. The selectivity values were assigned <1.01 if no chiral separations were observed, requiring a longer time than the 2 h analysis. Only *trans-*chrysanthemic acid methyl ester was analyzed for a longer time because some peak broadenings were recognized at 100 °C on the Alfa Dex 120 column. The calculated value was proven to be 1.007 at 100 °C, calculated from the data that were measured in the 60–80 °C range. The *cis*-chrysanthemic acid methyl ester showed also some chiral selectivity at low temperatures (1.014 at 80 °C). The free deltamethrinic acids demonstrated too low volatility and highly polar characteristics; therefore, they were not analyzed in free acid forms. The values of <1.01 without an asterisk show that no sign of the separations was recognized at 100 °C.

On every occasion, the free acidic forms (R_2_: H) show higher selectivity values than their methyl ester derivatives at the same temperature. On the other hand, the methyl ester forms gave better peak shapes (Figure 5) than the free acids (Figure 6). The perfect peak shape can be important in trace chiral impurity measurements because the tailings of the major peak of the free acids can cover the peak of other trace enantiomers.

It seems that the hydrogen bond donor ability of the well-positioned acid group of the analytes improves many of their chiral recognitions on several occasions. It is likely that the electrostatic interaction between the H in the acid groups and the H acceptors (the lone electrons of oxygen) of the selectors interact with each other, improving the chiral selectivity.

The permethyl alfa cyclodextrin has a chiral recognition feature towards only half of the tested pyrethroic acids. Chiral resolution was not recognized for the tested chrysanthemic compounds. The permethrinic acids are well suited for chiral recognition of permethyl α-CD mostly in underivatized forms of the acids. The *cis* isomers show much higher chiral selectivity than the *trans* isomers in α-CD selectors. The (+) isomers eluted before the (−) isomers on every occasion using the Alfa Dex 120 column.

The beta ring-sized selectors proved to be the best for chiral separations of the tested pyrethroic acids because it is an appropriate size to separate every tested compound (Table 1). The highest selectivity value (1.284) was obtained for *cis*-permethrinic acid on the Chirasil-Dex, which contained the chemically bonded permethyl β-CD in the silicone matrix. The Cydex-B stationary phase, a mixture of permethyl β-CD, was also successfully tested for the separation of pyrethroic acids. The results showed the same tendency of chiral selectivity for pyrethroic acids as the Chirasil-Dex did. However, the selectivity values were lower (generally 2–5%) on the Cydex-B than on the Chirasil-Dex. The differences in the selectivity values originated from the different silicon matrices and higher selector contents of the Chirasil-Dex. Namely, the more polar silicone matrix of the Cydex-B generates more intensive non-chiral interactions among the analytes and stationary phase than was observed on the Chirasil-Dex. The intensive achiral interactions decrease the chiral selectivity values because they are competitors for the interaction points of the analytes.

The enantiomers and *cis/trans* isomers of *cis*-chrysanthemic acids were excellently separated. The peaks of *cis* isomers surrounded the peaks of *trans* isomers, which were also well separated (Figure 7).

The chiral selectivity increases in the order of methyl, chlorine, and bromine substituents (R_1_) of vinyl groups. The hydrogen bond acceptor properties of the substituents possibly play an important role in their chiral selectivity. The *cis* isomers show higher selectivity than their *trans* isomers on the Chirasil-Dex. It is likely that the rigid cyclopropane ring structure of the pyrethroic acids fits well into the cavity of the permethyl β-CD. This inclusion process offers good chiral recognition for pyrethroic acids. The (+) isomers eluted before the (−) isomers on every occasion using the permethyl β-CD selector.

The gamma ring size of permethyl cyclodextrin can cause chiral separation in almost half of the tested compounds. The chrysanthemic acids (R_1_ = methyl) are well suited to the chiral recognition feature of the Gamma Dex 120 stationary phase. It was able to separate three of the four tested chrysanthemic enantiomeric pairs. Only the *trans*-permethrinic acid (R_1_: Cl,) showed some chiral selectivity from the non-methyl vinyl substituted molecules. The *trans* isomers showed much higher chiral selectivity than the *cis* isomers with the permethyl γ-CD selector. Only the enantiomers of the *trans*-chrysanthemic acid methyl ester were separated from the tested methyl ester derivatives with this selector. The higher selectivity values of chrysanthemic acid suggest that the substituents of the vinyl group (R_1_) have a key role in the chiral recognition processes. It is possible that the methyl substitutions are well suited to the inclusion properties of permethyl γ-CD. This chiral recognition mechanism of permethyl γ-CD is likely to be different from the chiral recognition mechanism of permethyl α-CD and permethyl β-CD. This was supported by the fact that the (−) isomers eluted first from the column that contained permethyl γ-CD (Figure 8), but the (+) isomers eluted first from the columns that contained permethylated α-CD and permethyl β-CD (Figure 5, Figure 6 and Figure 7).

A series of experiments were carried out (Table 2) to determine the best ester forms of various pyrethroic acids and establish the structure–chiral selectivity relationships [17]. The Chirasil-Dex was used for measurements, as this stationary phase and various combinations of analytes showed high selectivity values.

The free acids showed the highest selectivity values, as described in the previous chapter. The methyl ester showed much higher chiral selectivity among the ester derivatives of *cis*-permethrinic acid. It is better to analyze free acids if their peak symmetry is good (e.g., chrysanthemic acids). On the other hand, if the free acids show significant tailings, it is better to use methyl ester derivatives of pyrethroic acids (e.g., deltamethrinic acids). The selectivity order of normal alkyl esters of *cis*-permethrinic acid is as follows: methyl > ethyl > propyl > butyl. The tendency is not clear with branched alkyl esters. The secondary butyl ester showed rather high chiral selectivity (1.034), but the isopropyl and tertiary butyl esters of the acids did not show any chiral selectivity. It is likely that the oxo group H bond acceptor feature plays a role in the chiral recognition features. The alkyl groups of ester derivatives have shielding effects, which increase with the length of the n-alkyl chain of the ester derivatives.

### 2.2. Results of Supercritical Fluid Chromatography (SFC) Measurements

SFC is an effective method for chiral separation [35,36]. Recently, packed column SFC is the most used method in chiral selective separation. The applied columns were used originally in HPLC practice [36]. Capillary column SFC is not a frequently used technique in chiral separation, due to instrumental difficulties. However, some effective chiral separations were carried out using cyclodextrin that contained selectors on open tubular columns [18,35]. The chemically bonded silicone stationary phase contains permethyl β-CD (Chirasil-Dex), which has also been proven to be an effective stationary phase for SFC [18]. The peak shapes of the enantiomer of *cis*-permethrinic acid were perfect even at low temperatures of 50 °C, achieving an alpha value of 1.06. It was observed that the chiral selectivity values of the enantiomers (e.g., *cis*-permethrinic acid) showed lower values using SFC than using GC on the same column and with the same temperature. The functional groups of analytes and selectors have solvent spheres that must break for the intimate interactions between the analyte and the selector in SFC. Activation energy is needed to break up the solvent spheres, which somewhat decreases the chiral selectivity values. On the other hand, a significant solvent sphere does not exist when using GC. In general, the SFC measurements show higher chiral selectivity values than HPLC measurements using the same columns and temperatures. The different solvent spheres may be one of the possible explanations for the lower chiral selectivity values in HPLC than in SFC. The denser solvent spheres of the mobile phase around the functional groups of analytes and selectors need higher activation energy in HPLC compared to SFC.

### 2.3. Results of Capillary Electrophoresis (CE) Measurements

Electrophoresis is not a chromatographic technique. However, chiral selective capillary electrophoretic techniques use chiral selectors, which function as a “pseudo stationary phase”; therefore, the system shows chromatographic characteristics [25,27,28,31]. The distribution phenomenon plays a key role in the chiral separations in CE.

The oppositely charged selectors and analytes can show a high-resolution value [25,31]. The analytes show opposite mobility directions in negatively charged free acid forms compared to their associated states with positively charged selectors. These opposite mobility directions result in pseudo elongation of the separation length of the column. Even infinite resolution can be managed between the members of enantiomeric pairs with the fine tuning of parameters with oppositely charged selectors and selectants [37].

The GC results suggested that the β ring size also has chiral selectivity features toward pyrethroic acids on CE. The pyrethroic acid enantiomers have been effectively separated with various derivatives of β-CD in capillary electrophoresis [19,20,21]. Their chiral separations were possible with several neutral and positively chargeable CD derivatives. The β-cyclodextrin derivatives show good chiral recognition features for ionized forms of pyrethroic acids [21].

A preliminary study [19] showed that permethyl monoamino β-cyclodextrin (PMMAβCD) proved to be a good separation agent for isomers of *cis*-permethrinic acids even in its partly ionized state, achieving a resolution value of 9.4 (Figure 9).

Systematic studies were launched to find the optimum conditions for the chiral separations of pyrethroic acids using the PMMAβCD selector [20]. Most effective chiral separations were achieved when both partners, the positively charged β-CD and negatively charged pyrethroic acids, were ionized. The pK_a_ values of these acids cover the range of 5.1 ± 0.4, and the PMMAβCD has a pK_a_ of 9.05; therefore, to obtain the optimum pH of the background electrolytes, the range of 3.5–9.5 pH was researched. The optimum value was then confirmed as a pH of 6.5. The *cis*-deltamethrinic acid showed an R_s_ value of 20.0 using 15 mM PMMAβCD at this pH [20]. The achieved resolution values of the tested pyrethroic acids are summarized using 15 mM PMMAβCD and 15 mM permethyl β-CD (TRIMEB) chiral selectors (Table 3). These high-resolution values decreased steeply following a change in the pH in both directions using PMMAβCD. However, some decreased chiral selectivity preserved the system, when only one partner was ionized.

The baseline separation of every tested chiral pyrethroic acid during one analytical test was achieved using 15 mM PMMAβCD, except for *trans-*chrysanthemic acid [20]. The *trans*-chrysanthemic acid enantiomers can also be baseline separated using 17.5 mM PMMAβCD. The PMMAβCD unified the following two advantageous features: the positive charge and high methyl substitution rate in the chiral separations of pyrethroic acids. The high methyl substitution rate of PMMAβCD offers good solubility for the rather apolar pyrethroic acids in the water-based buffer. The good solubility of pyrethroic acids results in high-efficiency values, because it prevents the efficiency decreasing adsorption of analytes on the column wall. An efficiency score of 475,000 theoretical plates was achieved for *trans*-deltamethrinic acid using a 50 cm column length at 6.5 pH [20].

It is noteworthy that the resolution values of enantiomers of *cis*-permethrinic acids increased monotonically in the range between pH 5 and 9 using neutral TRIMEB, as the TRIMEB did not change its ionized state at high pH levels [21].The maximum resolution value of *cis*-permethrinic acid enantiomers, was recorded in the function of the concentration of TRIMEB. The 12.5 mMol concentration of the TRIMEB showed a maximum value of 2.6 at a pH of 7, in accordance with the Wrenn theory [38].

The chiral selectivity values were higher for *cis* isomers than for *trans* isomers. The selectivity order was as follows: deltamethrinic acid > permethrinic acid > chrysanthemic acids. In every case, the (−) isomers of pyrethroic acids migrated before the (+) isomers under the capillary electrophoretic conditions. The opposite migration orders were observed to that which was measured for the elution order in GC using β-sized selectors. The chiral recognition processes were likely to be the same in GC and CE using β-sized selectors. The more strongly bonded isomers migrated first when the EOF moved toward the cathodic direction, and the acidic selectants dissolved into the background buffer in CE. On the other hand, the less strongly bonded isomers eluted first in GC.

The enantiomers of *cis*-chrysanthemic acid were also separated under similar conditions as presented in Table 3, with the following additional selectors: monoamino β-cyclodextrin (R_s_: 2.57), dimethyl β-cyclodextrin, (R_s_: 1.93) and hydroxypropyl β-cyclodextrin (R_s_: 1.67) [20].

Different hydroxy amino-substituted β-CD derivatives produced better then baseline separations for *cis*-permethrinic acids and *cis*-deltametrinic acids [39].

Further experiments were carried out to solve the one-step analysis of diastereomeric salt resolution processes using TRIMEB as a chiral selector [21]. The unique feature of capillary electrophoreses makes the determination of acidic and basic compounds possible without any adsorption difficulties during analysis, even at trace levels. When the electrophoretic flow is sufficiently high, the cations and anions migrate towards the cathodic end of the capillary. The simultaneous determination of the enantiomeric ratio and composition of diastereomeric salt mixtures was evaluated and validated.

The well-chosen parameters are appropriate for the determination of the result of a diastereomer salt resolution process even on a short capillary column, as presented in Figure 10. The neutral environment (pH 7) of the background electrolyte was utilized to maintain the ionized states of the resolving agent and resolved *cis*-permethrinic acid. The migration times of 1-phenylethylamine and separated *cis*-permethrinic acid enantiomers were different than those for the EOF (Figure 10).

The randomly methylated-(6)-monoamino-deoxy-β-cyclodextrin also proved to be useful for the evaluation of the diastereomer salt resolution of pyrethroic acids.

No experiments were performed using ester forms of pyrethroic acids in CE. The ionic pyrethroic acids fit the chiral recognition features of derivatized β-CDs better than their ester forms in CE. The free acids, mostly in their ionized state, showed high chiral selectivity and perfect peak shapes.

## 3. Experimental Section

The tested pyrethroic acids, including chrysanthemic acids, (C_10_H_16_O_2_), permethrinic acids (C_9_H_12_Cl_2_O_2_), and deltamethrinic acids (C_9_H_12_Br_2_O_2_), were products of Ciba-Geigy (Basel, Switzerland). The following solvents and reagents used were products of Merck (Darmstadt, Germany): hexane, ethyl acetate, diethyl ether, BF_3_ etherate, methanol, ethanol, n-propanol, i-propanol, n-butanol, s-butanol, t-butanol, boric acid, phosphoric acid, acetic acid, NaOH, and HCl. Permethyl monoamino β-cyclodextrin (PMMAβCD), monoamino β-cyclodextrin (MAβCD), permethyl β-cyclodextrin (TRIMEB), dimethyl β-cyclodextrin (DIMEB) and hydroxypropyl β-cyclodextrin (HP βCD) were products of Cyclolab (Budapest, Hungary).

The chiral selective stationary phases used were the following: in-house-made Chirasil-Dex (silicone polymer substituted with anchored permethyl β-CD) [18], Cydex-B (mixture of silicone polymer and permethyl β-CD, (SGE, Melbourne, Australia), Alfa Dex 120 (mixture of silicone polymer and permethyl α-CD, Merck Life Science Kft, Budapest, Hungary), and Gamma Dex (mixture of silicone polymer and permethyl γ-CD, Merck Life Science Kft, Budapest, Hungary).

The chemicals of the background electrolytes of CE measurements were boric, acetic, and phosphoric acids (Britton Robinson buffer).

Carlo Erba Mega GC (Carlo Erba, Milan, Italy) and Shimadzu GCMS-QP5000 GC/MS instruments (Shimadzu, Kyoto, Japan) were used for the GC measurements [17,18]. An in-house-built SFC instrument was applied, combining a Carlo Erba Fractovap (Carlo Erba, Milan, Italy) and an Isco µLC 500 (ISCO, Belgium, Drogenbos) [18]. The capillary electrophoresis measurements were carried out on a Hewlett Packard 3DCE system (Hewlett Packard, Palo Alto, USA) with a diode array UV detector (202 and 220 nm) at 25 °C. Anuncoated silica-fused capillary column (58.5 cm × 50 µm i.d.) was applied [19,20]. The background electrolytes (BGE) consisted of 40 mM boric, acetic, and phosphoric acid buffers in the ratio of 1:2:2 (Britton-Robinson). The exact pH values of BGEs were adjusted with 0.1 M NaOH solution.

The syntheses of ester derivatives of pyrethroic acids were carried out using the reaction of BF_3_ etherate with various alcohols at 80 °C for two hours [40].

Every tested molecule was separated at least at 3 temperatures to measure or calculate the chiral selectivity values at 100 °C.

## 4. Conclusions

The enantiomeric separations of pyrethroic acids were accomplished with various gas chromatographic and capillary electrophoretic methods. The permethyl β-cyclodextrin (TRIMEB) selectors were the most effective in GC, but permethyl alfa- and gamma-cyclodextrin showed some chiral selectivity for pyrethroic acids. The most selective forms of the pyrethroic acids were the underivatized, free acid forms of pyrethroic acids, which were followed by the methyl ester derivatives. The selectivity values showed the following tendency: deltamethrinic acids > permethrinic acids > chrysanthemic acids, using beta ring-sized derivatives. The permethyl γ-CD showed a different separation mechanism than the other two ring-sized selectors under GC conditions. The elution orders of isomers were as follows: (+) first for alfa and beta derivatives, but (−) first for gamma-containing selectors in GC.

Capillary column SFC is also an appropriate technique for separating pyrethroic acid derivatives in cyclodextrins that contain stationary phases.

Several positively charged and neutral beta-cyclodextrin derivatives proved to be good resolving agents for pyrethroic acids. The pyrethroic acids show high chiral selectivity in their ionized forms. The permethyl monoamino β-cyclodextrin (PMMAβCD) positively chargeable selectors were the most effective for this task. The CE experiments showed maximum resolution values in the function of pH and the concentration of the selectors. The best selector was permethyl monoamino β-cyclodextrin (PMMAβCD) for the separation of pyrethroic acid enantiomers using CE, but other CD derivatives could also successfully separate pyrethroic acids+−.

The high chiral recognition properties of pyrethroic acids are partly due to their rigid cyclopropane structure. The H-bond donor and acceptor properties of the analytes improved the chiral recognition. Our findings and conclusions concerning the chiral recognition mechanisms may be useful in finding appropriate selectors for other chiral separations.

## Figures and Tables

**Figure 1 molecules-27-08718-f001:**
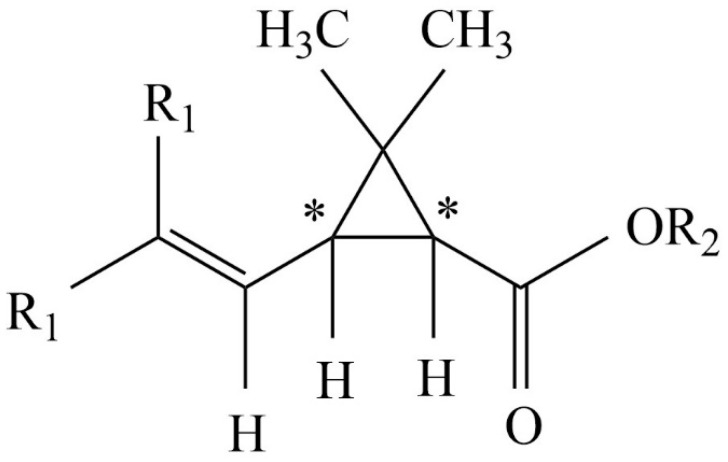
The general structure of the tested pyrethroic acids. The chiral carbon atoms are indicated with * symbols. The R_1_ can be methyl (chrysanthemic acid), chlorine (permethrinic acid), or bromine (deltamethrinic acid). The R_2_ can be hydrogen (free acid) or alkyl substituent (ester).

**Figure 2 molecules-27-08718-f002:**
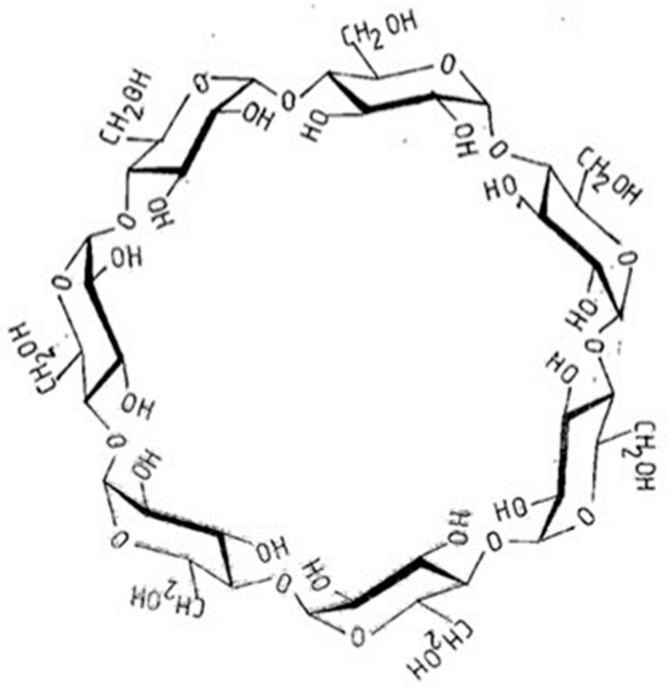
The structure of the β-cyclodextrin. The central cavity of the molecule can include guest molecules [28].

**Figure 3 molecules-27-08718-f003:**
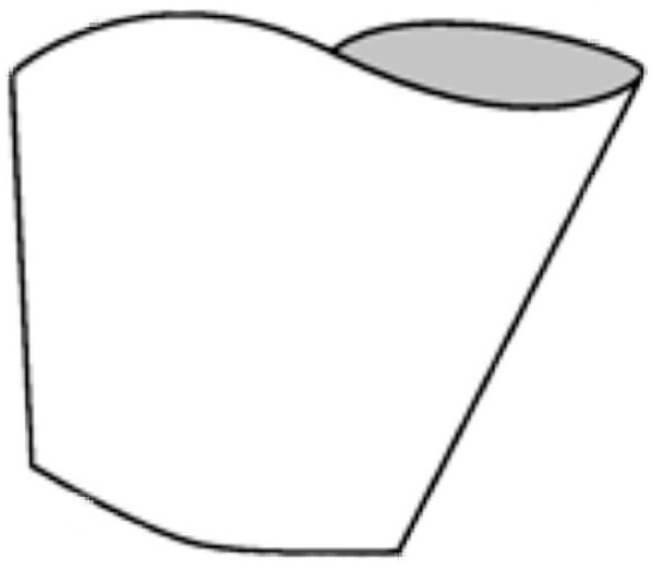
The real shape of derivatives of cyclodextrins is a twisted truncated cone, with various substituents, bond lengths, and directions around the chiral carbon centers.

**Figure 4 molecules-27-08718-f004:**
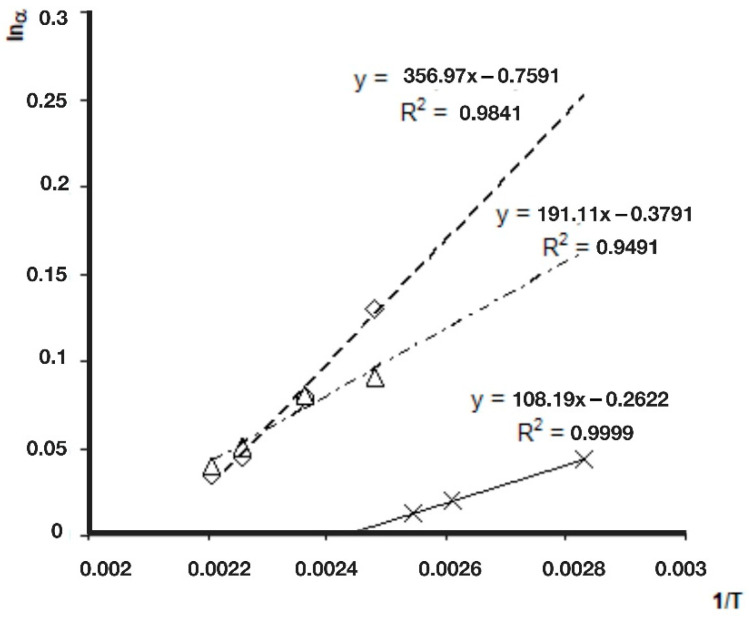
The natural logarithm of chiral selectivity values (lnα) of permethrinic acids in the function of inverse absolute temperatures (1/T). Symbols: ◊, *cis*-permethrinic acid; ∆, *trans*-permethrinic acid; x, *cis*-permethrinic acid methyl ester. No chiral separations were recognized for *trans*-permethrinic acid methyl ester, even at 80 °C. Conditions: instrument, Shimadzu GC/MS-QP5000 GC/MS; column, 25 m × 0.22 mm; stationary phase, Cydex-B (0.25 μm); carrier, He (50 cm/s).

**Figure 5 molecules-27-08718-f005:**
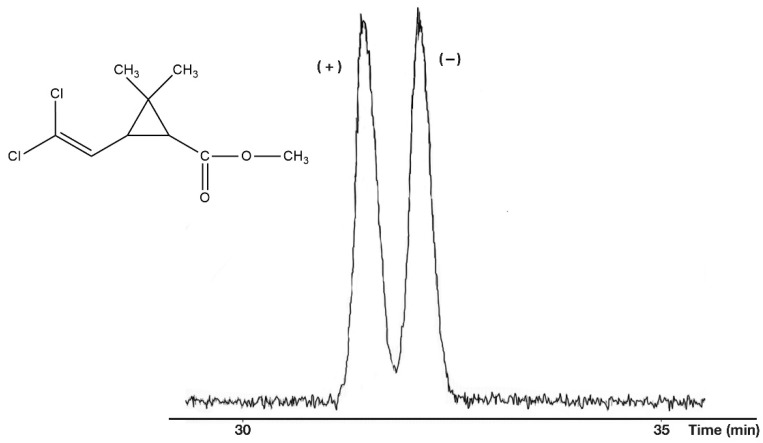
Chiral separation of *cis*-permethrinic acid methyl ester. Conditions: instrument, Shimadzu GC/MS-QP5000 GC/MS; column, 25 m × 0.22 mm; stationary phase, Cydex-B (0.25 μm); carrier, He (50 cm/s); temperature, 110 °C.

**Figure 6 molecules-27-08718-f006:**
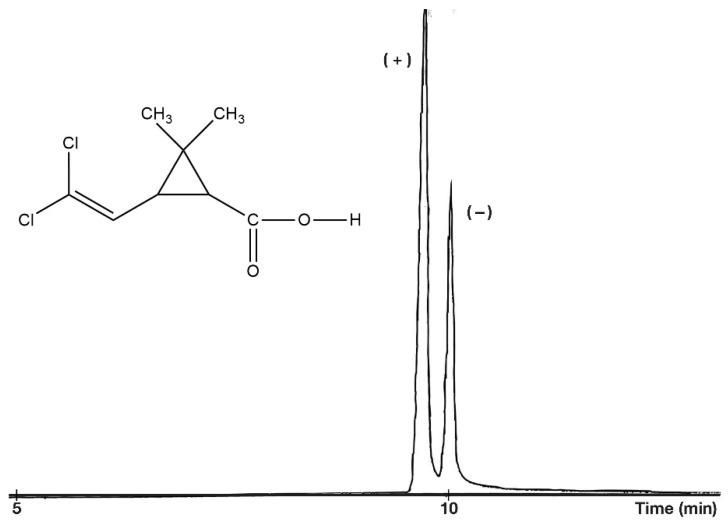
Chiral separation of *trans*-permethrinic acid methyl ester. Conditions: instrument, Shimadzu GC/MS-QP5000 GC/MS; column, 25 m × 0.22 mm; stationary phase, Cydex-B (0.25 μm); carrier, He (50 cm/s); temperature, 160 °C.

**Figure 7 molecules-27-08718-f007:**
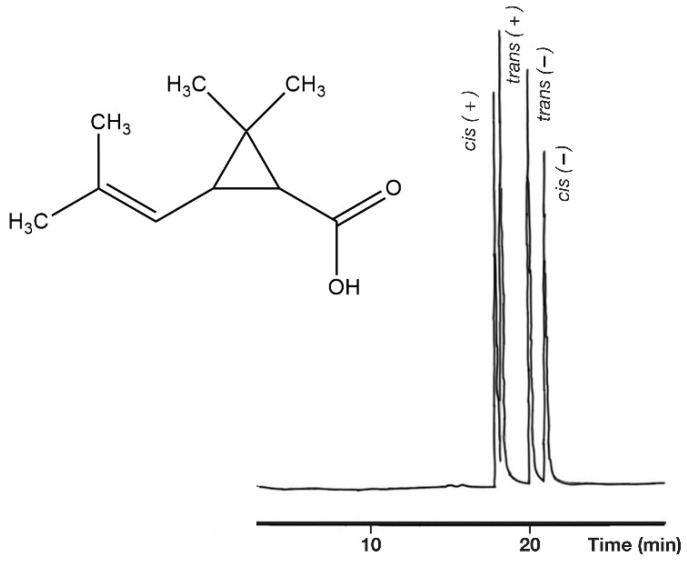
Chiral and *cis*-*trans* separations of chrysanthemic acid isomers as free acids. The peaks of *trans* isomers elute between the peaks of *cis* isomers. Conditions: instrument, Shimadzu GC/MS-QP5000 GC/MS; column, 25 m × 0.22 mm; stationary phase, Cydex-B (0.25 µm); carrier, He (50 cm/s); temperature, 135 °C.

**Figure 8 molecules-27-08718-f008:**
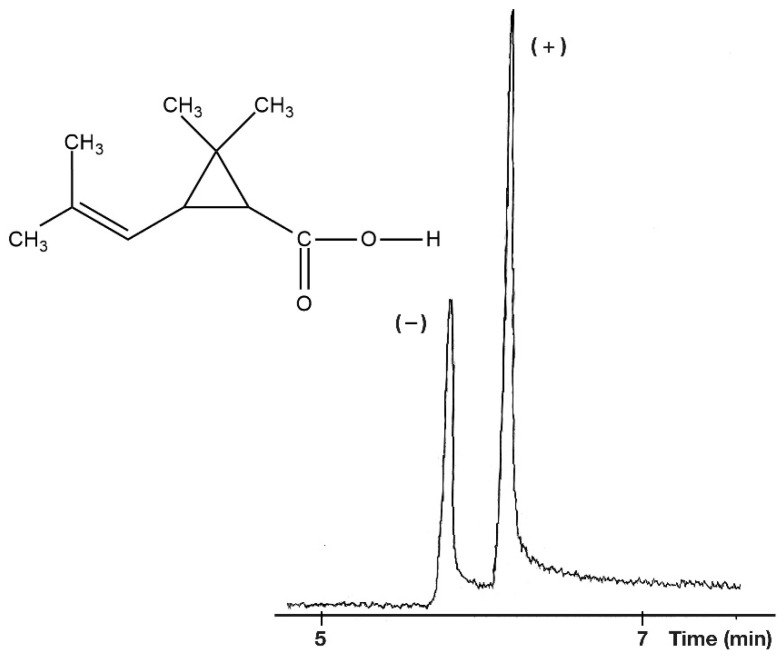
Separation of *trans*-chrysanthemic acid enantiomers with permethyl γ-CD selector. Conditions: instrument, Shimadzu GC/MS-QP5000 GC/MS; column, 25 m × 0.22 mm; stationary phase, Gamma Dex 120 (0.25 µm); carrier, He (50 cm/s); temperature, 150 °C.

**Figure 9 molecules-27-08718-f009:**
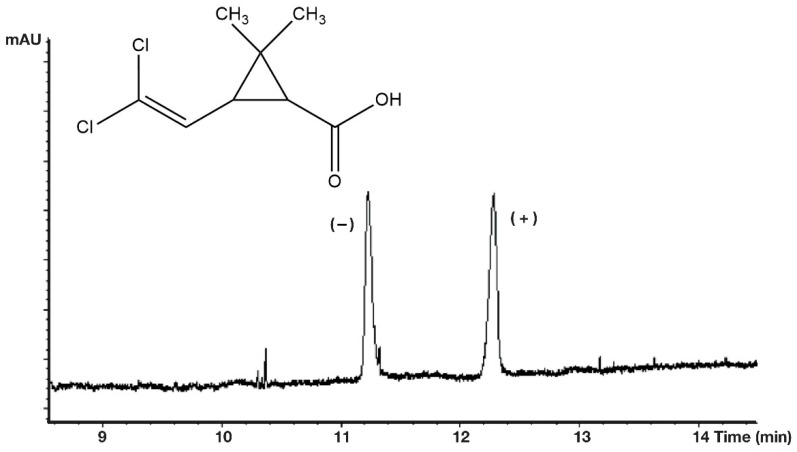
Chiral separation of *cis*-permethrinic acid enantiomers. Conditions: instrument, 50/58.5 cm silica-fused capillary; buffer, 50 mM phosphate buffer, pH 4.5; potential, 30 kV; temp, 25 °C; UV, 214 nm; selector, 10 mM PMMAβCD [19].

**Figure 10 molecules-27-08718-f010:**
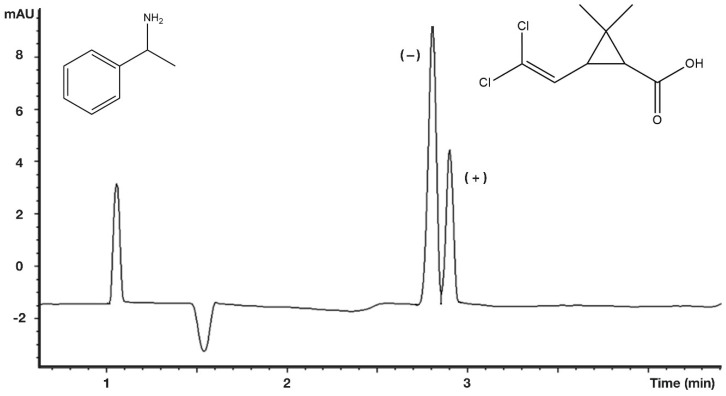
Electropherogram of raffinate of a diastereomeric salt resolution. Conditions: instrument, 33.5/25 cm × 50 µm column, FSOT; voltage, 20 kV; detection, 210 nm; temperature, 25 °C; buffer, 50 mM BRB, pH 7.0; chiral selector, 10.0 mM TRIMEB.

**Table 1 molecules-27-08718-t001:** The chiral selectivity values of pyrethroic acids for different ring-sized permethyl cyclodextrins containing stationary phases at 100 °C.

Structure of Tested Compounds			Permethylated Cyclodextrin Selectors			
R_2_	R_1_	*cis/trans*	Alfa Dex	CYDEX-B	Chirasil-Dex	Gamma Dex
Chrysanthemic acid						
H	Me	*cis*	<1.01	1.248	1.275	1.079
H	Me	*trans*	1.017	1.12	1.153	1.084
Me	Me	*cis*	<1.01	<1.01 *	1.013	<1.01
Me	Me	*trans*	<1.01 *	1.01	1.01	1.019
Permethrinic acid						
H	Cl	*cis*	1.224	1.219	1.284	<1.01
H	Cl	*trans*	1.14	1.142	1.194	1.095
Me	Cl	*cis*	1.024	1.028	1.043	<1.01
Me	Cl	*trans*	<1.01	<1.01	1.01	<1.01
Deltamethrinic acid						
Me	Br	*cis*	<1.01	1.038	1.046	<1.01
Me	Br	*trans*	<1.01	1.037	1.040	<1.01

R_1_, substituents of vinyl groups; R_2_, substituents of carboxyl groups (Figure 1); * shows some selectivity at lower temperatures.

**Table 2 molecules-27-08718-t002:** The selectivity values of *cis*-permethrinic acids (R_2_: Cl) in various derivative forms on the Chirasil-Dex at 100 °C.

Substituents of Carboxyl Function (R_2_)	Chiral Selectivity Value
Methyl ester	1.043
Ethyl ester	<1.023
Propyl ester	1.019
Isopropyl ester	<1.01
Butyl ester	1.014
Secondary butyl ester	1.034
Tertiary butyl ester	<1.01

**Table 3 molecules-27-08718-t003:** The resolution values of the tested pyrethroic acids using permethyl monoamino β-cyclodextrin (PMMAβCD) and permethyl β-cyclodextrin (TRIMEB) chiral selectors at pH 6.5.

Compounds	Resolution Values (R_s_)	
	15 mM PMMAβCD	15 mM TRIMEB
*trans*-deltamethrinic acid	11.56	3.24
*cis*-deltamethrinic acid	20	5.64
*trans*-permethrinic acid	6.62	1.69
*cis*-permethrinic acid	17.23	2.37
*trans*-chrysanthemic acid	1.08	<0.5
*cis*-chrysanthemic acid	8.5	1.56

Conditions: instrument, column with 50 cm effective length × 0.050 mm i.d., uncoated silica-fused capillary column; background buffer, pH 6.5 (40 mM boric, acetic, and phosphoric acid buffers, 1:2:2); potential, 30 kV; detection, 220 nm; temperature, 20 °C.

## Data Availability

The data presented in this study are available on request from the corresponding author.

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
