# Peer review of "Chiral Separations of Pyrethroic Acids Using Cyclodextrin Selectors"

_molecules, 2022, doi:10.3390/molecules27248718_

Round 1

Reviewer 1 Report

This work by Juvancz tackles an important aspect in separation science, by reporting chiral separation of pyrethroic acids, synthetic insecticide molecules, and an interesting model molecule for chiral separation studies. While applications of cyclodextrins for separation isn’t a novel finding, this work has laid out issues of enantioselective separation in a sound manner with the data presented being technically consistent also.   

This reviewer recommends publication of this submission with a few minor suggestions listed below. 

1.     Please revisit all text and revise as necessary. Writing ‘…in line of…’ is not a correct usage, rather ‘…in line with…’ is. Likewise using ‘…only limited extent.’ isn’t correct. These typos while minor in nature are distracting. Authors must change the sentence: ‘In this study, every tested enantiomer of pyrethroic acids were separated somehow under GC conditions’ to something along the lines of ‘In this study, every tested enantiomer of pyrethroic acids could be successfully separated using modular GC conditions’

2.     Can authors clarify why free acidic forms led in better selectivity results than methylated analogues? Is it due to electrostatically favorable interaction? To this note, authors’ clarification about methylated substituents leading in qualitatively better separation as a result of underlying peak symmetry is an important one. 

3.     While temperature-dependent study is relevant, there isn’t a consistent comparison between geometric isomers. Is it safe to assume that at 80 C there was no detectable separation for cis-chrysanthemic acid methyl ester. 

4.     Authors have studied carboxyl substituents for relative chiral selectivity studies, although it is not clear if their loading/extent of substitution is comparable in their studies and/or ligand normalized?

5.     Based on the results obtained for methyl, chlorine, and bromine substitution showing selectivity decrease in that order is indeed a reasonable observation based on hydrogen acceptor properties. Authors observation of PMMAβCD positively charged cyclodextrin analogue showing highest selectivity in their ionic forms is an important observation so is the better separation results obtained under super critical conditions.

Author Response

Dear Dr. Reviewer,

Thank you very much for your corrections.

Here are my responses:

Reviewer’s 1 note

  1. Please revisit all text and revise as necessary. Writing ‘…in line of…’ is not a correct usage, rather ‘…in line with…’ is. Likewise using ‘…only limited extent.’ isn’t correct. These typos while minor in nature are distracting. Authors must change the sentence: ‘In this study, every tested enantiomer of pyrethroic acids were separated somehow under GC conditions’ to something along the lines of ‘In this study, every tested enantiomer of pyrethroic acids could be successfully separated using modular GC conditions’.

Response

The reviewer is right our English is poor. We made the requested corrections. The manuscript was also corrected by a scientific expert who’s English much better than us.

Reviewer’s 1 note

  1. Can authors clarify why free acidic forms led in better selectivity results than methylated analogues? Is it due to electrostatically favorable interaction? To this note, authors’ clarification about methylated substituents leading in qualitatively better separation as a result of underlying peak symmetry is an important one.

Response

It seems that the hydrogen bond donor ability of the well positioned acid group of the analytes improves a lot of their chiral recognitions on several occasions. Probably the electrostatic interaction among the H of acid groups and the H acceptors (lone electrons of oxygens) of selectors interact each other improving the chiral selectivity. (after figure 6, page 10.)

It is better to analyse free acids if their peak symmetry is good (e.g., chrysanthemic acids). On the other hand, if the free acids show significant tailings, it is better to use methyl ester derivatives of pyrethroic acids (e.g. deltamethrinic acids). (top of page 13)

Reviewer’s 1 note

  1. While temperature-dependent study is relevant, there isn’t a consistent comparison between geometric isomers. Is it safe to assume that at 80 C there was no detectable separation for cis-chrysanthemic acid methyl ester.

Response

The values of < 1.01 without asterisk show that, no sign of the separations was recognized at 100° C. (bottom of page 8.)

Reviewer’s 1 note

  1. Authors have studied carboxyl substituents for relative chiral selectivity studies, although it is not clear if their loading/extent of substitution is comparable in their studies and/or ligand normalized?

Response

We did not make detailed studies for the carboxyl substitutions vs. loading capacity. We took care for the peak shapes only. The high methylation degree of selectors improved significantly the loadability of system, but this phenomenon was not studied in detailed.

Reviewer’s 1 note

  1. Based on the results obtained for methyl, chlorine, and bromine substitution showing selectivity decrease in that order is indeed a reasonable observation based on hydrogen acceptor properties. Authors observation of PMMAβCD positively charged cyclodextrin analogue showing highest selectivity in their ionic forms is an important observation so is the better separation results obtained under super critical conditions.

Response

No response was necessary.

Reviewer 2 Report

The manuscript by Zoltán Juvancz and co-authors reports on the chiral separation of phyretroic acids using cyclodextrin selectors, specifically, the authors tried derivatives of cyclodextrins to separate phyretroic acids under different conditions. The results sound interesting at the first glance. I would recommend publication after the authors consider the questions as follows:

1, the authors tried to justify the importance of chiral separation based on the effects of chirality on the compounds’ properties, which is well-known for college students who learnt the chirality of compounds. In my opinion, this is not enough, the authors should discuss the state-of-the-art in chirality separation of the targeted phyretroic acids, mention the pros/cons of the existing options, and then justify their motivation for this work.

2, the effects of different cyclodextrin derivatives are discussed on the functional groups, however, on a very superficial level, I believe the molecular recognition/interaction mechanism should be carefully analyzed, better with the support of theoretical calculations.

3, the chiral separation results should be compiled with results achieved from the best techniques to make a straightforward comparison of their new achievements. 

Author Response

Dear Dr. reviewer,

Thank you very much for your corections of  our manuscript.

Here are  my responses:

Reviewer’s 2 note

1, the authors tried to justify the importance of chiral separation based on the effects of chirality on the compounds’ properties, which is well-known for college students who learnt the chirality of compounds. In my opinion, this is not enough, the authors should discuss the state-of-the-art in chirality separation of the targeted phyretroic acids, mention the pros/cons of the existing options, and then justify their motivation for this work.

Response

The following sentence was attached:

Applying chiral pure agrochemicals becomes more and more accepted, because the members of enantiomer can have different activity in their uses and toxic effect in the nature. Several cases their different effects were recognized (e.g., pyrethrins, conocanozole, lindane, phenoxy propionic acids metaxilal, metolachlor etc.). (end of 1.1 chapter, page 2):

The structures, biological effects and analysis of the pyrethroids were discussed in detailed with their references.

The following sentence was added to the text:

Several chiral separations, however, remain unsolved in the field and production of pyrethroids. (end of 1.2 chapter, page 3):

We also emphasized that we try to establish structure- chiral selectivity relationships using cyclodextrin selectors, which can use other chiral compounds too.

Reviewer’s 2 note

2, the effects of different cyclodextrin derivatives are discussed on the functional groups, however, on a very superficial level, I believe the molecular recognition/interaction mechanism should be carefully analyzed, better with the support of theoretical calculations.

Response

The reviewer is right, we did not perform exact calculations, but our goal was only to establish trends and not to provide thermodynamic data. To gain exact chiral recognition data is rather difficult and frequently doubtful. The cyclodextrins have several thousand conformers, with very similar sterical and energetical properties (Frank Kobor, Klaus Angermund, Gerhard Schomburg, Molecular modeling experiments on chiral recognition in GC with specially derivatized cyclodextrins as selectors, Journal of High Resolution Chromatography 16(1993):299 – 311. It is very hard to select from the very similar conformers, which is responsible for the chiral recognitions. The literature shows several examples for the multimodal characters for the cyclodextrins. The retention order of enantiomers of lactones can be reversal along their homolog series (Carlo Bicchi et al. GC separation of the enantiomers of γ‐ and δ‐lactones on a mixture of 2,6‐dimethyl‐3‐trifluoroacetyl‐γ‐cyclodextrin and OV‐1701, Journal of High Resolution Chromatography 14(2005):701 – 704). The elution order can change according to derivate of analytes hydroxyl vs. acetate, (Juvancz, Z., Kiss, V., Schindler, J. et al. Use of Achiral Derivatization to Increase Selectivity and Reverse the Order of Elution of Enantiomers on Chirasil-Dex. Chromatographia 60 (Suppl 1), S161–S163 (2004).

Reviewer’s 2 note

3, the chiral separation results should be compiled with results achieved from the best techniques to make a straightforward comparison of their new achievements.

Response

The  following sentence was added:

The best selector is permethyl monoamino β-cyclodextrin (PMMAβCD) for the separation of pyrethroic acid enantiomers using CE method. (end of conclusion, ( end of chapter 5, page 17).

Best regards

Prof. Zoltán Juvancz

Round 2

Reviewer 2 Report

The authors properly resolved my previous comments. I support publication in its current form.